# An End-to-End Chromosome-Image Screening Framework for Radiation Biodosimetry

**Taeyoung Bak**[1]                                                TAEYOUNG22@UNIST.AC.KR
**Wol Soon Jo**[2]                                                   SAILORJO@DIRAMS.RE.KR
**Soo Kyung Jeong**[2]                                                SOO87@DIRAMS.RE.KR
**Su Jung Oh**[2]                                                  OSJ10050@DIRAMS.RE.KR
**Jimin Lee**[1,3]                                                   JIMINLEE@UNIST.AC.KR

[1] *Artificial Intelligence Graduate School, Ulsan National Institute of Science and Technology (UNIST), Ulsan, Republic of Korea*

[2] *Dongnam Institute of Radiological & Medical Sciences, 40 Jwadong-gil, Gijang-gun, Busan 46033, Republic of Korea*

[3] *Department of Nuclear Engineering, Ulsan National Institute of Science and Technology, Ulsan, Republic of Korea*

## Abstract

Radiation biodosimetry relies on chromosome-aberration analysis in blood lymphocytes, but manual scoring of dicentric chromosomes and translocations remains slow and expert-dependent. Prior studies have demonstrated automated dicentric-based biodosimetry, while translocation or FISH-based biodosimetry remains important for retrospective assessment. Deep-learning-based cytogenetic image analysis has also been demonstrated in fluorescence *mFISH* segmentation, and modern object detection has been applied to chromosomal-aberration analysis in a biodosimetry-adjacent setting. However, integrated modern image-based joint screening across both axes remains limited. We present an end-to-end chromosome-image screening framework that automates this pre-screening step with RT-DETR. Using a 15,450-image subset from a total of 18,818 images, we simplify the label space to `translocation(tr)`, `dicentric chromosome(dic)`, and `chromosome(chr)`. On the test split, AP50 reaches 0.758 for `dic` and 0.738 for `tr`. These results indicate meaningful localization of abnormality candidates in real data, although severe class imbalance, simplified labels, and scorable single-cell identification remain major bottlenecks.

**Keywords:** biodosimetry, chromosome aberration, dicentric chromosome, translocation, object detection

## 1. Introduction

Biological dosimetry estimates radiation exposure using biological changes observed after irradiation. Among the most widely used cytogenetic approaches are dicentric chromosome analysis and translocation analysis in blood lymphocytes. Dicentric chromosomes are useful for acute exposure assessment, whereas translocations are useful for retrospective dose assessment (Herate and Sabatier, 2020; Tucker and Luckinbill, 2011).

Reviewers must repeatedly determine whether an image corresponds to a scorable single cell and whether dicentric or translocation-related abnormalities are present. This workflow is labor-intensive (Wong et al., 2013), and scorable-cell screening itself remains a practical

bottleneck even before aberration scoring begins (Liu et al., 2017). These constraints motivate an AI-assisted screening step that localizes candidate abnormalities before downstream review.

Prior work already shows meaningful progress toward automated dicentric biodosimetry, including early machine-learning-based discrimination, automated dose estimation, and deep-learning-based dicentric analysis (Li et al., 2016, 2019; Jeong et al., 2022; Kim et al., 2023). Translocation or FISH-based biodosimetry is also established as an important retrospective setting (Tucker, 2001; Gregoire et al., 2018). In addition, deep-learning-based cytogenetic image analysis has been explored in *mFISH* segmentation (Pardo et al., 2018), and object detection has been applied to broader chromosomal-aberration analysis (Belyaev et al., 2026). To our knowledge, integrated image-based automation that jointly screens dicentric-related and translocation-related abnormalities within one workflow remains limited. We therefore present a framework that automates this screening stage as an initial module toward broader radiation biodosimetry.

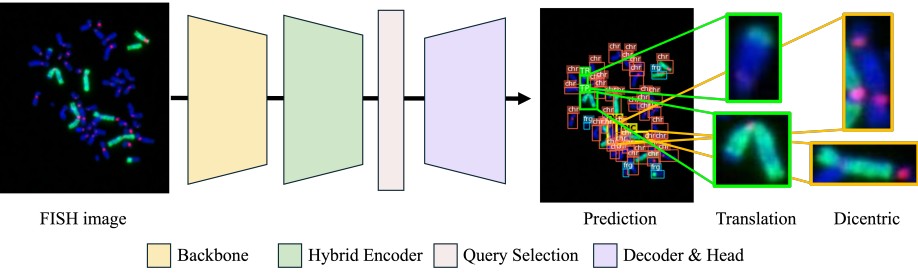

FISH image · Prediction · Translation · Dicentric

Backbone · Hybrid Encoder · Query Selection · Decoder & Head

Figure 1: Overview of the proposed end-to-end chromosome-image screening framework. The figure shows a simplified RT-DETR-based screening pipeline from a single chromosome image to localized predictions for `dic`, `tr`, and `chr`. See Zhao et al. (Zhao et al., 2024) for full RT-DETR architectural details.

## 2. Methods

**Dataset.** The dataset consists of 18,818 fluorescence-stained chromosome images acquired under 2, 3, 4, and 5 Gy conditions. All images were annotated by experts using a biologically fine-grained schema including target chromosomes, non-target chromosomes, dicentric-related labels, translocation-related labels, fragments, rings, insertions, and truncations. For feasibility testing, we simplify the labels so that dicentric-related labels map to `dic`, translocation-related labels map to `tr`, and all remaining chromosome labels map to `chr`. The dataset is severely imbalanced, with `tr` and `dic` cases accounting for less than 5% of the total.

**Experimental setup.** One of the main challenges is severe imbalance. We therefore use only images containing at least one `tr` or `dic` label, yielding a subset of 15,450 images split into training, validation, and test sets with an 8:1:1 ratio. We adopt RT-DETR (Zhao

Table 1: Class-wise detection performance on the test split.

| class | gt count | pred count | AP | AP50 | AP75 |
|-------|----------|------------|-------|-------|-------|
| dic | 2328 | 2334 | 0.576 | 0.758 | 0.660 |
| tr | 1221 | 1312 | 0.517 | 0.738 | 0.585 |
| chr | 70218 | 70359 | 0.821 | 0.956 | 0.912 |

et al., 2024) for end-to-end inference without non-maximum suppression and efficient hybrid encoding. We use focal loss for more stable learning under the imbalanced class distribution.

## 3. Experiments

**Quantitative results.** The class-wise detection performance on the test split is shown in Table 1. On the test split, AP50 reaches 0.758 for `dic` and 0.738 for `tr`. For both classes, mean AP remains lower than AP50, indicating room for improvement under stricter localization and class discrimination. The similar scales of `pred_count` and `gt_count` further suggest that the model is operating at an appropriate detection scale rather than trivially over- or under-predicting.

**Limitations and next steps.** Three limitations are clear from the current results. *(i) Class imbalance.* Because `dic` and `tr` have far fewer samples than `chr`, performance on the rare abnormality classes remains lower; data augmentation may help mitigate this. *(ii) Simplified label space.* Since the model works under this reduced label scheme, future studies should expand toward a richer label space that better reflects the original biological meaning. *(iii) Scorable single-cell identification.* Meaningful biodosimetry depends not only on chromosome-type classification, but also on whether the analyzed region corresponds to one scorable single cell (Liu et al., 2017). The current experiment focuses on joint abnormality screening, so scorable single-cell assessment, validation across dose groups, and the connection from image-level predictions to downstream biodosimetry aggregation remain to be addressed.

## 4. Conclusion

We present an initial framework for automated screening of dicentric- and translocation-related chromosome abnormalities from real annotated chromosome-image data, positioned toward automated radiation biodosimetry. Quantitative results on the filtered test subset show strong performance for the majority class `chr` and meaningful detection capability for the rarer classes `dic` and `tr`. These findings suggest that AI-assisted joint chromosome-aberration screening can serve as a feasible early step toward radiation biodosimetry, while class imbalance, label-space simplification, scorable single-cell assessment, and downstream biodosimetry integration remain open problems.

## Acknowledgments

This study was supported by a grant from the Korean Government (MSIT) to the Dongnam Institute of Radiological and Medical Sciences (DIRAMS; grant number 50491-2026).

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
