# OpenReview forum: "An End-to-End Chromosome-Image Screening Framework for Radiation Biodosimetry"
_MIDL.io/2026/Short_Papers — MIDL 2026 - Short Papers Poster_

### Official Review · Reviewer_KHrD · 2026-05-03

**Rating:** 4
**Confidence:** 4

**Review:**

This paper presents a clear and well-motivated application of deep learning to automate chromosome abnormality screening for radiation biodosimetry, using an RT-DETR-based object detection framework. The quality and clarity are good for a short paper, with a straightforward experimental setup and transparent discussion of limitations, though the methodological novelty is limited as it mainly applies an existing model. The work is significant from an application perspective, as it explores a unified approach to detecting both dicentric chromosomes and translocations, which aligns with practical workflows. Strengths include the use of real annotated data and reasonable detection performance under class imbalance, while weaknesses include simplified labels, dataset filtering bias, and limited experimental depth. Overall, it is a solid and meaningful preliminary study with room for further development.

**Summary:**

This paper addresses the question of whether an end-to-end deep learning framework can automatically pre-screen chromosome images for radiation biodosimetry by jointly detecting dicentric chromosomes and translocations. To answer this, the authors design a simplified object detection pipeline based on RT-DETR, reformulating the biologically complex annotation space into three categories (dic, tr, chr) and training on a filtered dataset enriched with abnormal cases. Experiments on a real cytogenetic dataset show that the model achieves reasonable detection performance (e.g., AP50 ≈ 0.76 for dicentric chromosomes), suggesting that abnormality localization is feasible even under severe class imbalance. The study is significant in that it explores a unified screening stage, which could reduce expert workload in biodosimetry pipelines, although several practical components remain unaddressed.

**Strengths:**

1. Addresses an important and practical problem in radiation biodosimetry by automating chromosome abnormality screening.
2. Proposes a unified framework that jointly detects dicentric chromosomes and translocations, aligning with real-world workflows.
3. Utilizes a relatively large, expert-annotated dataset, increasing the practical relevance of the study.

**Weaknesses:**

1. Limited methodological novelty, as it mainly applies an existing RT-DETR model without significant innovation.
2. Simplified label space reduces biological detail and may limit downstream applicability.
3. Dataset filtering (only abnormal images) introduces potential bias and reduces realism for deployment.

**Justification Of Rating:**

This submission fits well within the goals of the MIDL short paper track. It presents a relevant application, uses real-world data, and demonstrates promising initial results. While the methodological contribution is modest and the evaluation is somewhat limited, the paper does not exhibit any critical flaws in its design or conclusions. The authors are transparent about the limitations, and the work can be viewed as a meaningful early step toward a more complete automated biodosimetry pipeline. Given the typically inclusive nature of the short paper track and the practical importance of the problem, I believe this paper meets the bar for acceptance.

---

### Decision · Program_Chairs · 2026-05-08

Accept (Poster)